# Antioxidant and Anti-Inflammatory Defenses in Huntington’s Disease: Roles of NRF2 and PGC-1α, and Therapeutic Strategies

**DOI:** 10.3390/life15040577

**Published:** 2025-04-01

**Authors:** Francesco D’Egidio, Elvira Qosja, Fabrizio Ammannito, Skender Topi, Michele d’Angelo, Annamaria Cimini, Vanessa Castelli

**Affiliations:** 1Department of Life, Health and Environmental Sciences, University of L’Aquila, 67100 L’Aquila, Italy; francesco.degidio@graduate.univaq.it (F.D.); fabrizio.ammannito@guest.univaq.it (F.A.);; 2Department of Clinical Disciplines, University ‘Alexander Xhuvani’ of Elbasan, 3001 Elbasan, Albania; elvira.qosja@uniel.edu.al (E.Q.); skender.topi@uniel.edu.al (S.T.); 3Sbarro Institute for Cancer Research and Molecular Medicine, Temple University, Philadelphia, PA 19122, USA

**Keywords:** Huntington’s disease, oxidative stress, ROS, neuroinflammation, NRF2, PGC-1α, PPARs, neurodegeneration, antioxidant therapy

## Abstract

Huntington’s disease (HD) is a detrimental neurodegenerative disease caused by the expansion of a CAG triplet in the *HTT* gene. This mutation leads to the production of mutant Huntingtin (Htt) protein with toxic gain-of-function. The mHtt is responsible in several ways for the establishment of an intricate pathogenetic scenario in affected cells, particularly in HD neurons. Among the features of HD, oxidative stress plays a relevant role in the progression of the disease at the cellular level. Mitochondrial dysfunction, bioenergetic deficits, Reactive Oxygen Species (ROS) production, neuroinflammation, and general reduction of antioxidant levels are all involved in the promotion of a toxic oxidative environment, eventually causing cell death. Nonetheless, neuronal cells exert antioxidant molecules to build up defense mechanisms. Key components of these defensive mechanisms are the nuclear factor erythroid 2-related factor 2 (NRF2) and peroxisome proliferator-activated receptor gamma coactivator-1 α (PGC-1α). Thus, this review aims to describe the involvement of oxidative stress in HD by exploring the roles of NRF2 and PGC-1α, crucial actors in this play. Finally, antioxidant therapeutic strategies targeting such markers are discussed.

## 1. Introduction

Huntington’s disease (HD) is a severe neurodegenerative condition caused by the expansion mutation of the glutamine-encoding Cytosine-Adenine-Guanine (CAG) trinucleotide in exon 1 of the Huntingtin gene (HTT). The HTT gene, located on chromosome 4p16, is responsible for the production of the Huntingtin (Htt) protein, which is involved in many relevant cellular processes [1]. In HD, an elongated mutant Htt (mHtt) is produced due to the expanded gene, causing a toxic gain of function of the physiological Htt with an extended polyglutamine (polyQ) tract. Indeed, this feature is considered among the major hallmarks of HD pathology. Moreover, the presence of toxic N-terminal fragments containing only exon 1, produced by proteolytic cleavage of mHtt or CAG length-dependent aberrant splicing, significantly contributes to HD pathology at the cellular level [2].

From a clinical perspective, HD manifests through motor, psychiatric, and cognitive symptoms, with loss of functional autonomy over time. These symptoms reflect the pathological loss of neurons in the central nervous system (CNS), mainly in striatal γ amino-butyric acid (GABA)-ergic medium spiny neurons (MSNs), which are the most susceptible to HD-related damage [3].

HD is an uncommon autosomal-dominant disorder affecting approximately 5–10 individuals per 100,000 across most regions of North and South America, Australia, and Europe, while its prevalence (0.5/100,000) is considerably lower in Asian and African populations [4]. Although juvenile-onset cases exist, HD typically emerges between 35 and 50 years of age, with life expectancy generally ranging from 15 to 20 years after symptoms onset. Interestingly, disease onset is inversely correlated with the number of repeats within the mutated HTT gene. Indeed, adult patients typically show 36–50 repeats compared to the 10–35 of healthy people, whereas juvenile-onset cases exceed 50. The longer the HTT gene expansion, the faster mHtt toxicity spreads, aggregating in neurons and driving neurodegeneration locally and beyond. Thus, a complex network of pathophysiological events takes place within cells and tissues, with mitochondrial dysfunction, oxidative stress, and neuroinflammation as shared mechanisms observed in both the CNS and the periphery [5,6]. For instance, oxidative stress involvement in HD is profound: increased levels of antioxidant enzymes such as catalase, superoxide dismutase (SOD), and glutathione peroxidases (GPx), which are products of the antioxidant response element (ARE) genes, have been detected in HD-affected brains [7]. Under physiological conditions, oxidative stress is controlled by functional mitochondria that produce ATP through oxidative phosphorylation while maintaining redox balance and calcium homeostasis. However, massive reactive oxygen species (ROS) accumulation in HD cells damages intracellular molecules. Nuclear factor erythroid 2-related factor 2 (NRF2), a basic leucine zipper (bZIP) transcription factor that binds to ARE sequences, orchestrates the expression of antioxidant, anti-inflammatory, and detoxifying genes [8]. For instance, NRF2 pharmacological activation in rodent HD models has been shown to increase antioxidant expressions and reduce ROS levels [9].

Similarly, peroxisome proliferator-activated receptor γ coactivator-1 α (PGC-1α), a transcription factor involved in the regulation of several energetic processes in the cells, is downregulated in HD. As PGC-1α plays a pivotal role in maintaining redox balance and mitochondrial homeostasis under normal state, its downregulation in HD cells leads to detrimental effects such as impaired mitochondrial biogenesis and maintenance [10]. Despite their well-documented individual roles in oxidative stress regulation and mitochondrial function, discussions on NRF2–PGC-1α crosstalk are still lacking. Thus, this review aims to describe the involvement of NRF2 and PGC-1α in HD pathology, their crosstalk, and their potential as therapeutic targets for antioxidant strategies in HD treatment.

## 2. HD Pathogenesis

The formation of toxic mHtt monomers promotes the development of oligomers, which act as precursors to fibrils accumulating in cytoplasm and the nucleus, compromising the protective functions of the normal Htt with organelles dysfunction and cell loss. Specifically, mHtt impairs Htt’s control over apoptotic pathway inhibition and the production of neurotrophins such as brain-derived neurotrophic factor (BDNF) [11]. In addition, toxic mHtt can affect transcriptional regulation by interacting with DNA regions of numerous genes through its expanded polyglutamine tract [12].

HD also interferes with protein degradation systems, including the ubiquitin–proteasome system (UPS) and autophagy, as demonstrated in both animal models and human tissues [13,14]. Starting from the expansion mutation, mHtt production and accumulation within neurons and beyond can trigger gliosis in the CNS. The activation states of glial cells initiate intracellular signaling pathways that alter gene expression enhancing cytokine release, downregulating astrocytic glutamate transporter 1 (GLT1), impairing Kir4.1 channels, reducing BDNF levels, and elevating both glutamate and cytokine release from astrocytes, ultimately increasing oxidative stress and mitochondrial dysfunction, ROS production, and neuroinflammation [15].

Given the role of the cited mechanisms in HD pathogenesis and the involvement of NRF2 and PGC-1α in these processes, the following section describes oxidative stress, mitochondrial dysfunction, and neuroinflammation in more detail.

### 2.1. Oxidative Stress and Mitochondrial Dysfunction

Mitochondrial dysfunction is a crucial aspect of HD pathogenesis, with mitochondrial DNA (mtDNA) abnormalities closely linked to striatal degeneration and impaired respiratory chain complex function. Studies have highlighted disrupted mitochondrial morphogenesis in HD brains, characterized by enhanced fission, reduced fusion, and overall mitochondrial depletion. Alterations in mitochondrial dynamics induced by mHtt contribute to excessive mitochondrial fragmentation [16]. In fact, the Htt protein plays an essential role in maintaining mitochondrial integrity and function, while mHtt disrupts mitochondrial inner membrane proteins, promoting neuronal dysfunction and cell death.

In HD-affected neurons, mHTT interacts with mitofusin 1 (Mfn1) and the mitochondrial fission GTPase dynamin-related protein 1 (Drp1), further reducing mitochondrial fusion and exacerbating mitochondrial fragmentation and neuronal damage. These pathological alterations are associated with reduced mtDNA copy number and increased mtDNA damage [16]. Similarly, mtDNA alterations are linked to altered dopamine metabolism––a neurotransmitter significantly implicated in HD pathogenesis.

Mitochondria play a crucial role in neurotransmitter synthesis; their impairment can lead to altered neurotransmission involving dopamine, acetylcholine, glutamate, and GABA [17]. For instance, glutamate synthesis occurs within astrocytic mitochondria, where energy-dependent processes such as ATP generation are crucial for its transfer from astrocytes to neurons and its storage in synaptic vesicles [18]. Mitochondrial dysfunction has been identified as a major contributor to glutamate-induced excitotoxicity, which compromises neuronal resistance by increasing mitochondrial and cytosolic Ca^2+^ accumulation, hallmarks of HD-dependent excitotoxic damage. Dysregulation of GABA-A receptors has also been related to altered mitochondrial membrane potential, intensifying neurodegeneration.

Moreover, the breakdown of antioxidant defenses under glutamate excitotoxic conditions amplifies oxidative stress, further impairing mitochondrial dynamics and accellerating disease progression [19]. Impaired mitochondrial axonal transport is another early contributor to HD progression [20]. In HD, mHtt interferes with biogenesis while disrupting both anterograde and retrograde transport governed by kinesin and dynein motor proteins. This interference results from mHtt binding directly to these motor proteins, impairing mitochondrial attachment, reducing transport efficiency, and inducing cell death through mitochondrial membrane damage [21]. Thus, mitochondrial dysfunction disrupts calcium homeostasis and intensifies ROS production, promoting oxidative stress and neuronal injury in HD. The interplay between mitochondrial dysfunction and oxidative stress is pivotal in the establishment of HD pathogenesis [22]. Researchers demonstrated a significant correlation between CAG repeat length and oxidative DNA damage [22]. Furthermore, mHtt has been shown to enhance oxidative stress in both neuronal and non-neuronal cells, thereby intensifying HD pathology.

Nuclear accumulation of mHTT is a hallmark of HD pathology, with oxidative stress implicated as a crucial factor for its establishment [23]. HD pathology is primarily characterized by increased oxidative stress and heightened vulnerability of the striatum to mHTT toxicity. Genetic studies revealed that mHTT expression induces transcriptional alterations, leading to enhanced oxidative stress and upregulation of antioxidant genes associated with the NRF2–ARE pathway, a compensatory mechanism aimed at mitigating oxidative damage [24].

Additionally, mHTT downregulates PGC-1α expression and activity––a crucial co-regulator of mitochondrial biogenesis and antioxidant enzyme synthesis––thereby increasing susceptibility to striatal degeneration and oxidative damage [25]. Collectively, the evidence underscores the critical role of mitochondrial dysfunction and oxidative stress in HD pathogenesis, while highlighting potential therapeutic targets aimed at preserving mitochondrial integrity and mitigating oxidative stress-driven neuronal damage.

### 2.2. Neuroinflammation

Concerning neuroinflammation, HD has been associated with elevated levels of inflammatory markers in both the central and peripheral nervous systems [15]. Research has indicated that disease progression correlates with increased pro-inflammatory cytokines and a decline in anti-inflammatory cytokines. Glial cells, which play a pivotal supportive role for neurons in the CNS, become activated in response to a pro-oxidant and pro-inflammatory environment [26,27]. Indeed, astrocyte reactivity has been observed in the early, pre-symptomatic stages of HD and correlates with disease progression [28]. In HD, astrocytes are predominantly neurotoxic, contributing to elevated pro-inflammatory cytokines and ROS at synapses, disrupted potassium balance, and impaired glutamate uptake, which results in membrane depolarization [29]. On the other hand, a few neuroprotective astrocytes resist and contribute to the antioxidant response facilitating the restoration of damaged neuronal circuits [30].

Similarly, microglial activation triggered by various pro-inflammatory signals leads to the secretion of both pro- and anti-inflammatory cytokines, such as interleukins (IL) 1β (IL-1β), IL-4, IL-6, and tumor necrosis factor α (TNF-α), as well as growth factors like transforming growth factor β (TGF-β), mannose receptor (CD206), and arginase 1 (Arg1) [27]. Evidence from post-mortem analyses of HD brains reveals accumulation of reactive microglia, particularly in the frontal cortex and striatum, regions closely associated with HD pathology [31]. Interestingly, HD-related neurodegeneration correlates with the density of activated microglia, underscoring their role in neuronal damage.

Mitochondrial dysfunction in microglial cells appears to impair IL-4-driven anti-inflammatory responses, further promoting inflammation [32,33]. This dysfunction may contribute to elevated pro-inflammatory mediator expression and neuronal injury. Interestingly, NRF2 has been recognized for its role in suppressing neuroinflammation by reducing inflammatory responses in neurons [8]. This protein is involved in the assembly of the inflammasome, a protein complex responsible for regulating inflammation. Specifically, NRF2 interacts with the nucleotide-binding oligomerization domain (NOD)-like receptor containing pyrin domain 3 (NLRP3), an inflammasome sensor protein, to downregulate inflammatory signaling. Through this mechanism, NRF2 appears to modulate neuroinflammation, contributing to the mitigation of HD symptoms [34]. Additionally, oxidative stress induced by activated microglia has been linked to neurotoxicity, partly due to increased ROS production, which further amplifies proinflammatory gene expression and microglial and astrocytic activation. In this context, researchers described the modulation of the pro-oxidant and pro-inflammatory environment by anti-inflammatory microglia via the PGC-1α pathway, suggesting its therapeutic potential in HD-driven neuroinflammation [35].

## 3. Role of NRF2 and PGC1α

After outlining HD and its key pathological features––oxidative stress and neuroinflammation––the focus now shifts to a detailed examination of NRF2 and PGC-1α, emphasizing their roles in both physiological and pathological conditions. The discussion concludes by exploring potential interactions between these two regulators, and therapeutic strategies targeting their pathways.

### 3.1. NRF2

NRF2 regulates the cellular response to oxidative stress by binding to ARE, which functions as a regulatory enhancer sequence located within the promoter regions of detoxifying genes, modulating the expression of genes involved in antioxidant defense and inflammation control [36]. Thus, activation of NRF2 signaling leads to the upregulation of antioxidant enzymes, including GPx, SOD, heme oxygenase-1 (HO-1), and NAD(P)H quinone oxidoreductase-1 (NQO1) (Figure 1). Furthermore, NRF2 induces anti-inflammatory mediators and protects against mitochondrial dysfunction, underscoring its crucial role in cellular resilience against oxidative stress and inflammation [37].

From a structural perspective, NRF2, a member of the cap-and-collar (CNC) bZIP transcription factor family, consists of seven functional domains, designated Neh1 through Neh7. The C-terminal region includes Neh1, Neh3, and Neh6 domains. Within Neh1, which harbors the CNC bZIP motif, NRF2 interacts with DNA by forming heterodimers with Maf proteins [38]. The Neh2 domain, located at the N-terminal, plays a pivotal role in cytoplasmic regulation by facilitating NRF2 binding to the Kelch-like ECH-associated protein-1 (Keap1) [39]. The Neh3 domain recruits chromodomain helicase DNA-binding protein-6 (CHD6), which modulates transcriptional activity, while the Neh6 domain contains DSGIS and DSAPGS motifs, which are essential for β-TrCP-mediated NRF2 degradation [40]. Additionally, the Neh7 domain interacts with the retinoic acid receptor (RAR), further regulating NRF2 activity [41].

Keap1 predominantly controls NRF2 stability under basal conditions by anchoring it in the cytoplasm. Two Keap1 molecules can bind a single NRF2 protein, promoting its ubiquitination and proteasomal degradation. Upon oxidative stress, NRF2 dissociates from Keap1, translocates into the nucleus, and accumulates at higher levels. This nuclear accumulation enhances NRF2 binding to AREs within the promoter regions of target genes involved in antioxidant defense, metabolism, and cellular homeostasis [41]. Nonetheless, other regulators are involved in NRF2 accumulation, such as p21; protein kinase C and p62, which interfere with KEAP1-NRF2 binding; hydrogen peroxide, which stimulate NRF2 translation; and HECT domain and ankyrin repeat, containing E3 ubiquitin protein ligase 1 (HACE1) tumor suppressor, which is essential for NRF2 activation under oxidative stress, as it promotes NRF2 protein synthesis, stabilization, and nuclear localization. Regarding the latter, researchers have recently described the deep connection between HACE1 expression and NRF2 signaling. Specifically, HACE1 positively regulates the NRF2-mediated antioxidant response using its ankyrin repeats and HECT domain, but independently of its E3 ligase activity. In HD contexts, knock-out models for HACE1 and NRF2 depicted the interconnection between these pathways, showing that absence of HACE1 impaired NRF2, while restoring HACE1 expression enhanced NRF2 activity and oxidative balance [42]. Building on NRF2’s pivotal role, various signaling pathways converge to modulate its activity and orchestrate cellular defense. For instance, the mitogen-activated protein kinase (MAPK) cascade, known for regulating cell proliferation, differentiation, and apoptosis, is activated by reactive oxygen species generated during calcium imbalance [43]. While MAPK signaling typically promotes pro-inflammatory responses, NRF2 activation counteracts this effect by upregulating antioxidant enzymes and suppressing inflammatory mediators, thereby restoring redox homeostasis [44,45,46]. In addition, AMP-activated protein kinase (AMPK) senses energy deficits and responds to metabolic stress by activating NRF2 [47]. Early AMPK activation reduces neuroinflammation and works synergistically with NRF2 to elevate antioxidant enzyme expression, thereby safeguarding neuronal function [48,49]. Concurrently, extracellular signal-regulated kinase (ERK) reinforces NRF2 activity, aiding recovery from DNA damage and excitotoxic events [50]. Similarly, the phosphoinositide 3-kinase (PI3K)/protein kinase B (Akt) pathway, crucial for cell survival and metabolic regulation, enhances NRF2 activity [51]. Stimulated by growth factors and hypoxic stress, Akt promotes NRF2 nuclear translocation and transcription of detoxifying genes, effectively mitigating oxidative damage [52,53,54]. This crosstalk exemplifies how NRF2 serves as a central mediator against cellular stress. Sirtuins (SIRTs) further refine this defense network. In both mice and humans, metabolic processes, including mitochondrial function, gluconeogenesis, and cell survival, are governed by sirtuins, a family of NAD^+^-dependent histone deacetylases. These enzymes orchestrate cellular energy production and overall metabolism and their pivotal role in metabolic regulation is unequivocally recognized [55]. SIRT1-mediated deacetylation stabilizes NRF2, increasing its capacity to drive antioxidant responses, while inhibition of SIRT2 has been linked to neuroprotection in experimental models [56,57,58]. Such modulation underscores NRF2’s role in preserving cellular integrity under oxidative challenges. The interplay extends to the mammalian target of rapamycin (mTOR) pathway, where reduced mTOR activity is associated with neurodegeneration [59]. NRF2 regulates mTOR expression via ARE, mitigating neuronal atrophy. Moreover, modulation of p53 in response to oxidative stress further emphasizes NRF2’s central role [60]. Indeed, lower stress conditions favor NRF2-driven defenses, while higher stress triggers p53-mediated apoptosis. Collectively, these interconnected pathways show NRF2 as the master regulator of cellular resilience against oxidative stress and inflammation. By integrating signals from MAPK, PI3K/Akt, SIRTs, AMPK, ERK, mTOR, and p53, NRF2 coordinates a comprehensive protective response that maintains neuronal viability and offers therapeutic targets for neurodegenerative disorders. Thus, the integrative network governed by NRF2 not only fortifies cellular defenses but also represents a promising focus for innovative therapeutic strategies.

### 3.2. PGC-1α

The brain’s high metabolic requirements, alongside the critical role of ATP production and mitochondrial integrity in neuronal function, have led to extensive investigations into PGC-1 coactivators as key regulators of energy metabolism. Expression studies have revealed abundant PGC-1α presence in several brain regions, including the cerebral cortex, hippocampus, striatum, thalamic nucleus, and substantia nigra [61]. PGC-1α serves as a versatile transcriptional coactivator, interacting with a broad range of transcription factors involved in various biological processes. These include mitochondrial biogenesis, oxidative phosphorylation (OXPHOS), antioxidant responses, adaptive thermogenesis, and metabolic regulation of glucose and fatty acids (Figure 2) [62]. Indeed, the PGC-1 family consists of proteins that are tightly regulated in response to various environmental signals and are essential in modulating signaling pathways to ensure proper cellular and systemic adaptations to changing conditions. PGC-1α, the most thoroughly studied member, was initially identified as a protein interacting with PPARγ in brown adipose tissue [63]. PGC-1β, which closely resembles PGC-1α, shares considerable sequence similarity, while the PGC-related coactivator (PRC) has more limited homology [64]. Regarding PGC-1α, it forms complexes with transcription factors such as NRF1, NRF2, nuclear receptors including PPARα, PPARδ, PPARγ, and estrogen-related receptor α (ERRα) [25]. These complexes regulate the expression of nuclear-encoded mitochondrial genes like cytochrome c (Cyt c), mitochondrial complexes I–V, and mitochondrial transcription factor A (TFAM), which are crucial for mitochondrial function. Mitochondrial defense against ROS is primarily mediated by antioxidant enzymes such as SOD2, which converts superoxide anions to hydrogen peroxide, along with peroxiredoxins (Prx3 and Prx5), mitochondrial thioredoxin (Trx2), and thioredoxin reductase (TrxR2) [65,66]. PGC-1α is recognized for suppressing oxidative stress by upregulating mitochondrial uncoupling proteins (UCPs) and antioxidant enzymes including SOD1, SOD2, and GPx-1. Regulatory promoter regions of SOD2, UCP-2, and Prx5 are directly bound by PGC-1α, establishing a central role in transcriptional control [67,68]. Additionally, while the nuclear SIRT1 controls PGC-1α activation status via deacetylation regulating the energy levels inside cells, mitochondria express SIRT3 under PGC-1α regulation, which deacetylates and activates SOD2, thereby reducing ROS and enhancing fatty acid oxidation [69]. Therefore, through the activation of mitochondrial gene networks, PGC-1α enhances fatty acid β-oxidation, the tricarboxylic acid cycle, and OXPHOS efficiency. Additionally, it promotes the expression of genes associated with ion transport, heme biosynthesis, and mitochondrial protein synthesis, while boosting overall respiratory capacity [70,71]. PGC-1α has also been associated with the maintenance of multiple neurotransmitter systems, including cholinergic, glutamatergic, dopaminergic, and GABAergic synapses [72,73,74]. A deficiency of PGC-1α within specific brain regions, particularly GABAergic neurons, has been linked to hyperactivity, impaired short-term habituation, and heightened startle responses [75]. Conversely, PGC-1α activation or overexpression has demonstrated neuroprotective effects by enhancing mitochondrial function, supporting neuronal survival, reducing neuroinflammation, and promoting protein clearance [35,76]. PGC-1α plays a multifaceted role in HD, impacting mitochondrial function, protein homeostasis, and neuronal integrity [10]. It regulates postnatal myelination by promoting myelin basic protein (MBP) expression and cholesterol synthesis [77]. Both adult HD models and PGC-1α knockout mice show decreased MBP levels and impaired myelination, while PGC-1α overexpression enhances MBP promoter activity, suggesting its critical role in myelin maintenance. At the organelle level, PGC-1α upregulation increases mitochondrial mass, promotes organelle fusion, and restores mitochondrial dynamics [78]. In HD mouse models, brown adipose tissue exhibits a significant reduction in functional mitochondria and a lower ATP/ADP ratio [79]. This mitochondrial dysfunction correlates with decreased expression of PGC-1α target genes involved in energy metabolism, indicating that impaired PGC-1α signaling may drive broader metabolic deficits in HD. On a molecular scale, PGC-1α stimulates transcription factor EB (TFEB), a key regulator of the autophagy-lysosome pathway, facilitating the clearance of mHtt, thereby supporting cellular proteostasis and reducing the accumulation of toxic aggregates commonly observed in HD [80,81]. Given its role in mitochondrial function, protein turnover, and myelination, enhancing PGC-1α activity has emerged as a promising therapeutic strategy for early intervention in HD, offering potential benefits for both neuronal maintenance and metabolic homeostasis.

### 3.3. Interaction Between NRF2 and PGC-1α

NRF2 accumulates in the cytoplasm bound to Keap1, a negative regulator. Upon exposure to oxidative stress, NRF2 dissociates from Keap1, translocates into the nucleus, and binds to regulatory enhancer sequence ARE in the promoter regions of various detoxifying genes (e.g., GPx, SOD, and NQO1) [7,24]. This nuclear accumulation of NRF2 not only upregulates antioxidant enzymes but also amplifies cellular response to inflammation, thus playing a central role in cellular resilience against stress. Interestingly, NRF2 activation is tightly regulated by various signaling pathways, one of the most crucial being AMPK [82]. As described by Joo and collaborators, AMPK phosphorylates NRF2 at Ser50, which in turn leads to the inactivation of glycogen synthase kinase 3β (GSK3β), which is overexpressed in oxidative stress and inflammation promoting NRF2 degradation, thereby facilitating NRF2’s translocation to the nucleus [82,83]. This process is essential for NRF2’s function as a transcriptional activator of antioxidant and cytoprotective genes. Moreover, various compounds such as berberine, fidarestat, and pterostilbene have been shown to activate NRF2 via AMPK signaling, highlighting the importance of this pathway in cellular redox regulation [84,85,86]. When AMPK is inhibited, NRF2 activation is abolished, underscoring the interdependent nature of AMPK and NRF2 signaling [87]. Beyond AMPK, NRF2 activity is intricately connected to PGC-1α (Figure 3). It was demonstrated that PGC-1α can promote the transcriptional activation of antioxidant genes via NRF2 [88]. Downregulation of PGC-1α expression in mice led to a significant reduction in the binding of NRF2 to specific genes like GCLC, affecting the levels of SOD2 and glutamate cysteine ligase (GCL) proteins. Further studies in PGC-1α-deficient mice showed disrupted NRF2-dependent mitochondrial biogenesis, pointing to a crucial role of PGC-1α in orchestrating NRF2 activation under stress conditions [89]. The interaction between NRF2 and PGC-1α is thought to be governed by the p38 MAPK pathway, which is positively regulated by PGC-1α and leads to GSK3β inactivation, further promoting NRF2 activation [90]. This interplay suggests that NRF2 and PGC-1α may form a feedback loop. NRF2 itself may regulate the expression of PGC-1α, enhancing mitochondrial function and antioxidative defense. Indeed, the PGC-1α gene promoter contains two ARE sequences (−1723 and −226) which are directly regulated by NRF2 [91]. Experiments involving NRF2 siRNA silencing or NRF2 knockout resulted in a decrease in mitochondrial biogenesis and down-regulation of PGC-1α expression across various cell types, including hepatocytes and skeletal muscles, further supporting the existence of NRF2-PGC-1α feedback regulation [92,93]. Simultaneous activation of the NRF2 and PGC-1α pathways may also be observed. For instance, the ERK1/2 pathway activate both NRF2 and PGC-1α via phosphorylation of liver kinase B1 (LKB1), which subsequently activates AMPK [94]. In the context of HD, NRF2 presents a potent avenue for therapeutic exploration. Indeed, NRF2 activation has been shown to protect neurons from oxidative stress and mitochondrial dysfunction, and overexpression of NRF2 offers neuroprotection from mHtt toxicity [9,95]. Conversely, deletion of NRF2 renders mice more susceptible to neurotoxins like malonate, a compound that mimic HD pathophysiology [96]. Alongside NRF2, PGC-1α levels are reduced in HD as previously described, further implicating mitochondrial dysfunction as a central component of HD pathology. Interestingly, direct interactions between PGC-1α and NRF2 may occur in a bidirectional manner to maintain mitochondrial resilience under stress.

## 4. Antioxidant Therapeutic Strategies Involving NRF2 and PGC-1α Signaling

### 4.1. Modulation of NRF2 as a Therapy for HD

Therapeutic interventions aimed at modulating the NRF2 pathway have attracted significant attention as potential strategies to combat HD. In various preclinical studies (reviewed below), neuroprotection has been achieved by enhancing NRF2 activity using a range of pharmacological agents that mitigate oxidative stress and inflammation. Animal models of HD have been instrumental in delineating the beneficial effects of these compounds, which include naturally occurring phytochemicals and synthetic compounds (Table 1). Phytochemicals have emerged as promising candidates in the therapeutic landscape of HD. Naringin, a flavanone abundant in citrus fruits able to promote AMPK phosphorylation and upregulation of Nrf2 expression, has been shown to alleviate inflammation and oxidative stress in male Wistar rats exposed to 3-nitropropionic acid (3-NPA), a mitochondrial inhibitor [97]. In these studies, the upregulation of the NRF2 pathway by naringin led to a reduction in neuroinflammatory markers and oxidative damage. Consistent results have been observed in PC12 cell cultures, where naringin’s activation of NRF2 contributed to a significant decrease in 3-NPA-induced neurotoxicity [53].

Protopanaxatriol, an active constituent derived from ginseng, has been similarly implicated in neuroprotection via the promotion of NRF2 nuclear translocation [98]. When tested in rat models of 3-NPA-induced neurodegeneration, protopanaxatriol activated the NRF2 pathway and was found to mitigate neuronal injury, further supporting the role of NRF2 modulation in countering HD pathology. Similarly, 6-shogaol, a compound contained in ginger able to modify multiple cysteine residues of Keap1 protein, recently showed remarkable effects in a 3-NPA-induced HD rat model [99]. Indeed, improved behavior and biochemical indices with restored levels of neurotransmitters and decreased neuroinflammatory molecules were achieved via NRF2 upregulation and NFkB-dependent inflammation modulation caused by 6-shogaol administration. Notably, the expression of the exon 1 in *HTT* is influenced by NRF2-associated mechanisms [100]. This finding has led researchers to explore compounds that activate NRF2, as a means of modulating HD-associated gene expression. Among these compounds, triterpenoids have been extensively investigated for their ability to upregulate NRF2 expression [2]. For instance, the triterpenoid derivative 2-Cyano-3,12-dioxo-olean-1,9-dien-28-oic acid-methyl-amide (CDDO-MA) exhibits excellent blood–brain barrier permeability and has demonstrated neuroprotective effects in 3-NPA neurotoxic models [101]. In these models, the activation of the NRF2 signaling pathway by CDDO-MA counteracted neuronal damage. In a complementary approach, transgenic mice expressing a 171-amino acid N-terminal fragment of human mHtt with 82 CAG repeats (N171-82Q), sufficient to cause striatal atrophy, were treated with compounds such as CDDO-ethyl amide (CDDO-EA) and CDDO-tri-fluoroethyl amide (CDDO-TFA) [95,102]. In these experiments, the transcription of NRF2-regulated genes was enhanced, which correlated with improved motor performance and increased neuronal survival in the striatum. Another recent study demonstrated that azilsartan (Azil), an angiotensin II type 1 receptor blocker able to downregulate Keap1, exhibited neuroprotective effects in a 3-NPA-induced neurotoxicity rat model [103]. Azil improved motor function, restored neurotransmitter balance, and reduced inflammation, oxidative stress, and apoptosis through modulation of NFkB and NRF2/Keap1 pathways, highlighting its therapeutic potential. Additional evidence supporting the neuroprotective role of NRF2 comes from investigations employing cysteamine [104]. In its reduced form, cysteamine exerts inhibitory effects on several enzymes that are implicated in neurodegenerative processes [105]. When administered in various HD in vitro models, cysteamine conferred protection by promoting NRF2 activation. This activation is achieved through alkylation and oxidation of cysteine residues on Keap1, thereby counteracting the deleterious effects of oxidative stress [104].

The therapeutic portfolio is further broadened by dimethyl fumarate (DMF), which has been recently evaluated in both the YAC128 and R6/2 mouse models of HD [9,106,107,108]. DMF treatment has been associated with improvements in muscle function, stabilization of body weight, and a marked reduction in neuronal loss in critical brain regions, including the cortex and striatum. These outcomes suggest that DMF’s activation of NRF2-dependent pathways––occurring via alkylation of Keap1 cysteine residues and nuclear exclusion of the ARE-site NRF2 competitor BACH1––contributes to enhanced neuronal survival and functional recovery. Even though preclinical studies are promising for these substances, there is a lack of robust clinical evidence. Among these antioxidant agents, DMF is the only approved neurodegenerative therapy (e.g., multiple sclerosis) that acts via NRF2 activation [109].

However, resveratrol and epigallocatechin gallate (EGCG) have recently demonstrated antioxidant effects via NRF2 in animal models, even though these models were for conditions different from HD [110]. Based on evidence of a resveratrol-dependent blockage of Keap1 and its neuroprotective effects showed in preclinical research using 3-NPA-induced models using male C57BL/6J mice, 3-NPA-treated rats, and N171-82Q transgenic mice, a clinical investigation evaluated resveratrol’s effect on caudate volume in HD patients (ID: NCT02336633). Similarly, a randomized, double-blind phase II trial assessed the safety, tolerability, and cognitive benefits of EGCG, a green tea polyphenol able to oxidize Keap1 cysteine residues and to downregulate BACH1, accelerating the NRF2-Keap1 complex disassociation and preventing BACH1-dependent ARE site occupation, in HD patients (ID: NCT01357681). Unfortunately, the results of both trials remain unpublished. Finally, cysteamine is currently being tested only in a dose-finding and tolerability trial with HD patients (ID: NCT02101957).

**Table 1 life-15-00577-t001:** Overview of the antioxidant therapeutic strategies involving NRF2 and PGC-1α signaling.

Therapeutic Strategy	HD Model	NRF2 Activation	PGC-1α Activation	Effects	Ref.
Naringin	3-NPA-stressed rats	V		Reduced neuroinflammatory markers and oxidative damage	[97]
	3-NPA-stressed PC12 cells	V		Reduced 3-NPA-induced neurotoxicity	[53]
Protopanaxatriol	3-NPA-stressed rats	V		Reduced neuronal injury	[98]
6-Shogaol	3-NPA-stressed rats	V		Improved behavior and biochemical indices with restored levels of neurotransmitters and decreased neuroinflammatory molecules	[99]
CDDO-MA	3-NPA-stressed rats	V		Reduced neuronal injury	[101]
CDDO-EA	N171-82Q mice	V		Improved motor performance and increased neuronal survival in striatum	[95]
CDDO-TFA	N171-82Q mice	V		Improved motor performance and increased neuronal survival in striatum	[95]
Azilsartan	3-NPA-stressed rats	V		Improved motor function, restored neurotransmitter balance, and reduced inflammation, oxidative stress, and apoptosis	[103]
Cysteamine	Mouse primary neurons and human iPSCs	V		Reduced oxidative stress and neuroprotection	[104]
	Human patients	V		(Ongoing study)	NCT02101957
DMF	YAC128 and R6/2 mice	V		Improvements in muscle function, stabilization of body weight, and a marked reduction in neuronal loss	[9]
Resveratrol	Human patients	V		(Unpublished results)	NCT02336633
	Striatal and cortical neurons from YAC128 mice, Human lymphoblasts, and YAC128 mice		V	Rescued mitochondrial membrane potential and respiratory activity with TFAM upregulation	[111]
EGCG	Human patients	V		(Unpublished results)	NCT01357681
PGC-1α overexpression	N171-82Q mice		V	Improved neurological function and TFEB-mediated eradication of mHtt aggregates in the brain	[80]
	R6/2 mice		V	Halted striatal degeneration	[112]
Fenofibrate	3-NPA-stressed rats		V	PPARγ-mediated enhanced mitochondrial oxidative phosphorylation and biogenesis.	[113]
	Human patients		V	Improved motor and cognitive symptoms	NCT03515213
Rolipram	Quinolinic acid-stressed rats		V	Antidepressant and neuroprotective effects	[114]
	R6/2 mice		V	Neuroprotective effects	[115]
	R6/2 mice		V	Improved motor functions and neuroprotection	[116]
GSK356278	Human patients		V	Able to reach the brain	NCT01602900
SIRT1 overexpression	N171-82Q mice		V	Improved metabolism and neuroprotection, decreased brain atrophy	[117]
	R6/2 mice		V	Improves survival, neuropathology and neurotrophins levels	[118]
Viniferin	Striatal cells and primary cortical neurons from HD mice, N63-148Q PC12, and N2A cells		V	SIRT3-dependent neuroprotection	[119]
Rosiglitazone	Striatal cells from HD mice		V	Reduced mitochondrial dysfunction and oxidative stress	[120]
Benzafibrate	R6/2 mice		V	Improved behavior and survival, decreased striatal atrophy and oxidative stress	[121]

### 4.2. Modulation of PGC-1α as a Therapy for HD

As mitochondrial dysfunction and oxidative stress remain central to HD progression, researchers explore strategies that enhance mitochondrial quality control as a means of neuroprotection. One promising avenue is the upregulation of PGC-1α (Table 1). Recent experimental models have demonstrated that elevating PGC-1α levels can not only restore energy production but also promote the clearance of toxic protein aggregates, thereby attenuating neuronal loss. Indeed, in an innovative genetic study, mice engineered to overexpress PGC-1α in a controlled, inducible manner exhibited a striking reduction in the accumulation of mHtt aggregates via activation of TFEB, which is crucial in the autophagy–lysosome system [80,81]. Activation of TFEB facilitates the degradation of misfolded proteins and damaged organelles, effectively resetting the intracellular protein quality control machinery [81]. Such clearance mechanisms are vital for maintaining neuronal integrity and mitigating the toxic effects of mutant proteins. Moreover, using a lentiviral vector to introduce PGC-1α into the striatum of R6/2 HD mice effectively halted local tissue degeneration, underscoring the promise of gene therapy strategies that enhance mitochondrial regulation [112]. Parallel to genetic approaches, several small molecule therapies are currently under investigation to indirectly boost PGC-1α activity. For instance, fenofibrate, a well-known agonist of PPARs, has been shown to modulate PGC-1α function in 3-NPA rats [113,122]. By binding to PPARs, fenofibrate initiates a cascade of transcriptional events that enhance mitochondrial oxidative phosphorylation and biogenesis. Recently, a six-month phase 2 clinical trial described the safety and efficacy of fenofibrate administration in HD patients and improved motor and cognitive symptoms compared to placebo with no adverse effect observed across the study and specific limitations described, reflecting the translational potential of targeting metabolic regulators to combat neurodegeneration (ID: NCT03515213). Natural compounds have also received considerable attention. Resveratrol, a polyphenol present in red wine and other sources, acts as a potent activator of SIRT1, which by deacetylating PGC-1α enhances its transcriptional effects on mitochondrial genes, leading to improved mitochondrial respiration and overall energy metabolism [123]. Mechanistically, resveratrol inhibits cAMP-degrading phosphodiesterases, thereby increasing intracellular cAMP levels, which in turn raise calcium concentrations and activate the CamKKβ-AMPK pathway. This cascade ultimately promotes the formation of NAD^+^ and further stimulates SIRT1 activity. Interestingly the neuronal inhibition of phosphodiesterase IV (PDE4), a resveratrol’s target, has emerged as a strategy to elevate cAMP levels, thereby indirectly boosting PGC-1α expression [124,125]. For instance, rolipram, a potent PDE4 inhibitor, has demonstrated both antidepressant and neuroprotective effects in preclinical and clinical HD models by increasing protein kinase A (PKA) activity [114,115,116,126]. However, despite its efficacy, rolipram’s narrow therapeutic window and gastrointestinal side effects have hindered its clinical adoption. Notably, GSK356278, a new PDE4 inhibitor, has been tested in a clinical setting addressing its ability to reach the brain in challenge with Rolipram in eight subjects with no significant safety concerns (ID: NCT01602900).

Nonetheless, preclinical data from models using 3-NPA suggest that resveratrol can ameliorate motor deficits, although its benefits in the striatum appear limited by suboptimal brain penetration [127,128,129]. Notably, resveratrol’s capacity to restore mitochondrial function extends beyond neuronal cultures. In lymphoblasts isolated from HD patients, treatment with resveratrol completely reinstated mitochondrial membrane potential and respiratory activity while upregulating mitochondrial TFAM [111]. Furthermore, enhanced SIRT1 levels correlate with improvements in motor coordination, a reduction in brain atrophy, and mitigation of mHtt–induced metabolic disturbances [111,117,118].

The therapeutic implications extend to the mitochondrial SIRT3, which is intimately involved in maintaining mitochondrial integrity. Located within the mitochondrial matrix, SIRT3 deacetylates components of the respiratory chain like Complex I and enhances fatty acid oxidation and antioxidant defenses by activating enzymes like SOD2 [119]. In HD cellular models, SIRT3 expression is diminished, yet pharmacological induction with viniferin is able to reverse these deficits, leading to neuroprotection [119]. Beyond direct modulation of PGC-1α, activation of nuclear receptors such as PPARs offers an alternative route to improving mitochondrial function via PGC-1α. In HD cells, activation of PPARγ with rosiglitazone has been shown to prevent calcium-induced mitochondrial dysfunction and reduce oxidative stress [120]. Additionally, bezafibrate, a pan-PPAR agonist, has recently been reported to elevate PGC-1α expression, ameliorate behavioral deficits, extend survival, reduce striatal atrophy, and diminish oxidative damage in the R6/2 mouse model [121]. While the potential risk of tumorigenesis linked to PPAR agonists remains a concern, the cumulative evidence suggests that modulation of the PGC-1α pathway, whether through direct overexpression or via upstream nuclear receptor activation, represents a multifaceted strategy to restore mitochondrial integrity, mitigate oxidative stress, and ultimately protect neurons in HD [130,131]. This innovative approach may offer a promising avenue for the development of effective therapies aimed at slowing disease progression and improving patient outcomes.

## 5. Discussion

Mitochondrial dysfunction, neuroinflammation, and oxidative stress are key hallmarks of HD, posing significant challenges for effective treatment. As therapy for HD remains challenging, targeting NRF2 offers a novel and promising intervention by reducing oxidative stress and preserving mitochondrial function. NRF2’s regulation of mitochondrial biogenesis and its feedback loop with PGC-1α ensure the maintenance of cellular homeostasis, which is compromised during the progression of these diseases. NRF2 modulation via natural phytochemicals (naringin, protopanaxatriol, 6-shogaol), synthetic triterpenoids (CDDO derivatives), angiotensin receptor blockers (azilsartan), cysteamine, and dimethyl fumarate (DMF) effectively reduced neuronal damage and improved motor function in preclinical models by enhancing antioxidant gene expression and suppressing inflammatory responses. Although clinical trials involving resveratrol, EGCG, and cysteamine have been initiated, their results remain unpublished or preliminary.

Concurrently, targeting mitochondrial dysfunction through PGC-1α activation has demonstrated beneficial effects on neuronal survival. Genetic induction of PGC-1α significantly reduced mHtt aggregation by promoting autophagy via TFEB activation. Pharmacological agents including fenofibrate, resveratrol, rolipram, rosiglitazone, and bezafibrate similarly improved mitochondrial function and reduced oxidative stress, mitigating neuronal loss and behavioral deficits in animal models. Clinical data support fenofibrate’s safety and potential therapeutic benefit, although challenges remain regarding optimal brain penetration and side effects of certain compounds. The combined focus on both NRF2 activation and simultaneous upregulation of mitochondrial functions via PGC-1α could substantially slow HD progression and improve patient outcomes.

Over the past decades, extensive studies have elucidated the multifaceted role of mitochondria in neuronal survival and degeneration. Mitochondria are not only the powerhouses of the cell but also play critical roles in programmed cell death. Their dynamic nature allows them to alter morphology, replicate, and even fuse or divide in response to metabolic demands or stress signals. Genetic mutations, aberrant mtDNA, and disrupted transcriptional regulation contribute to a decline in mitochondrial performance. Such dysfunction is now recognized as a central player in HD pathophysiology. Recent preclinical investigations have provided robust evidence that targeting mitochondrial deficiencies can be a viable therapeutic avenue. Specifically, strategies aimed at correcting mitochondrial dynamics and intracellular trafficking are gaining attention. However, clinical trials targeting these pathways have often been unsuccessful due to incomplete understanding of the underlying molecular networks, heterogeneous patient populations, and the absence of reliable biomarkers for therapeutic efficacy. Indeed, bridging the gap between preclinical discoveries and clinical application requires rigorous validation of safety, robust proof of efficacy, and an in-depth understanding of HD complexity. Moreover, challenges in translating preclinical successes into clinical benefits extend beyond molecular intricacies. The blood–brain barrier (BBB) remains a formidable obstacle for drug delivery, limiting the effective concentrations of many promising compounds in the CNS [132].

Innovative drug delivery systems, including nanoparticle-based carriers and viral vectors for gene therapy, may provide the means to overcome these obstacles, as outlined by recent research [133,134]. Additionally, the integration of advanced imaging and biochemical biomarkers could facilitate early detection of therapeutic effects, enabling more refined clinical trial designs [135,136]. Notably, while much attention has been focused on pharmacological approaches, non-pharmacological interventions such as physical exercise, dietary modifications, and controlled exposure to mild stressors (like cold or hypoxia) have also been shown to upregulate NRF2 and PGC-1α and enhance mitochondrial function [137,138]. These lifestyle interventions could complement pharmacotherapy by providing a holistic strategy to maintaining cellular energy homeostasis and reducing oxidative stress. Combining such non-pharmacological approaches with targeted molecular therapies may yield synergistic effects, offering new hope for HD patients.

Furthermore, the complexity of HD necessitates a multifaceted approach that goes beyond single-target interventions. The interplay between various cellular processes—ranging from protein quality control and autophagy to neuroinflammatory signaling—suggests that a combinatorial therapy approach may be more effective. In summary, a comprehensive strategy that integrates NRF2 activation with enhanced PGC-1α–mediated mitochondrial biogenesis represents a promising frontier in HD therapy. By restoring mitochondrial function, reducing oxidative stress, and improving metabolic efficiency, this approach has the potential to mitigate neuronal loss and slow disease progression. Looking forward, advancements in drug delivery, biomarker discovery, and combined therapeutic modalities will be crucial. These innovations not only promise to refine our understanding of HD’s complex molecular landscape but also hold the potential to translate into transformative clinical therapies that enhance quality of life and extend survival in patients afflicted with neurodegenerative disorders.

In particular, developing effective antioxidant and anti-inflammatory therapies for HD by targeting NRF2 and PGC-1α requires a systematic, multi-phase strategy. Firstly, preclinical studies must identify candidate compounds that can synergistically activate NRF2 and enhance PGC-1α expression, thereby boosting cellular defense against oxidative stress and inflammation in relevant preclinical models, as establishing robust biomarkers to assess redox balance and mitochondrial function is crucial for patient stratification. Early-phase clinical trials should then focus on evaluating the safety, tolerability, and pharmacokinetic profiles of these combination therapies. Following this, phase II studies must provide proof-of-concept through randomized, controlled designs, measuring both biochemical markers and clinical outcomes. Large-scale phase III trials are essential to confirm efficacy and refine dosing regimens via adaptive trial designs. Lastly, early regulatory engagement and ongoing post-marketing surveillance will ensure long-term safety and therapeutic benefit. Nonetheless, while preclinical results offer promising insights into therapeutic potential, cautious interpretation is crucial. Animal models and in vitro studies do not fully replicate human biology, often leading to discrepancies between laboratory findings and clinical outcomes. Historical challenges in translating preclinical successes into effective treatments highlight the importance of realistic expectations. Thus, acknowledging limitations, advocating rigorous clinical testing, and carefully balancing optimism with scientific realism are essential for effectively bridging the gap between preclinical promise and clinical applicability.

## 6. Conclusions

In conclusion, NRF2 and PGC-1α integrate multiple cellular processes critical for stress response, mitochondrial biogenesis, and antioxidant and neuroinflammatory defenses. Indeed, the crosstalk between these two factors is essential for maintaining cellular homeostasis during oxidative stress and neuroinflammation. Their synergistic work in protecting against the cited toxic events positions them as key targets for therapeutic tools in HD and other neurodegenerative disorders.

## Figures and Tables

**Figure 1 life-15-00577-f001:**
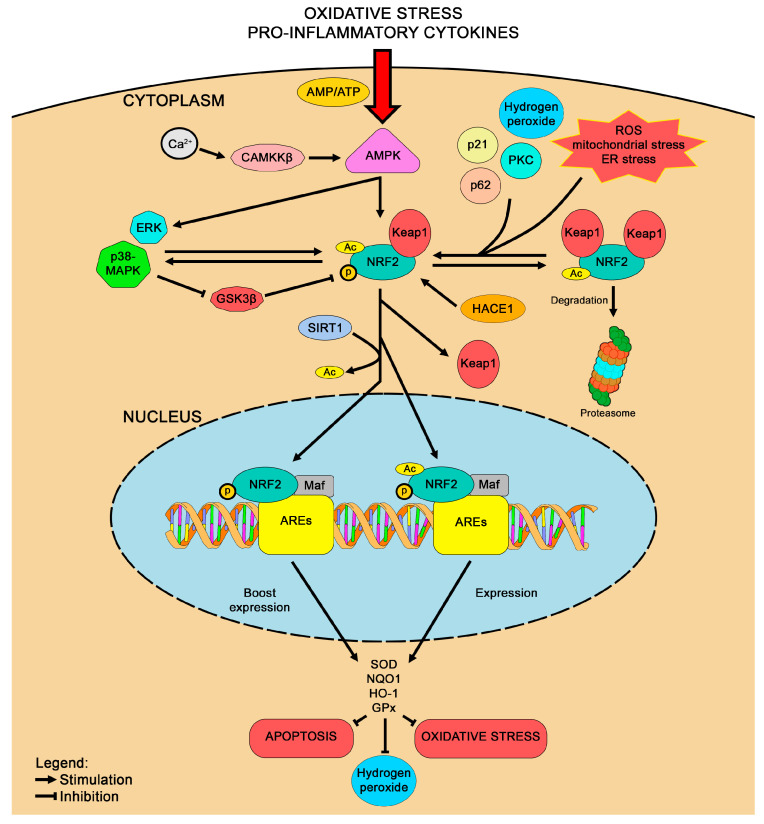
Molecular mechanisms underlying the antioxidant effects of NRF2 signaling.

**Figure 2 life-15-00577-f002:**
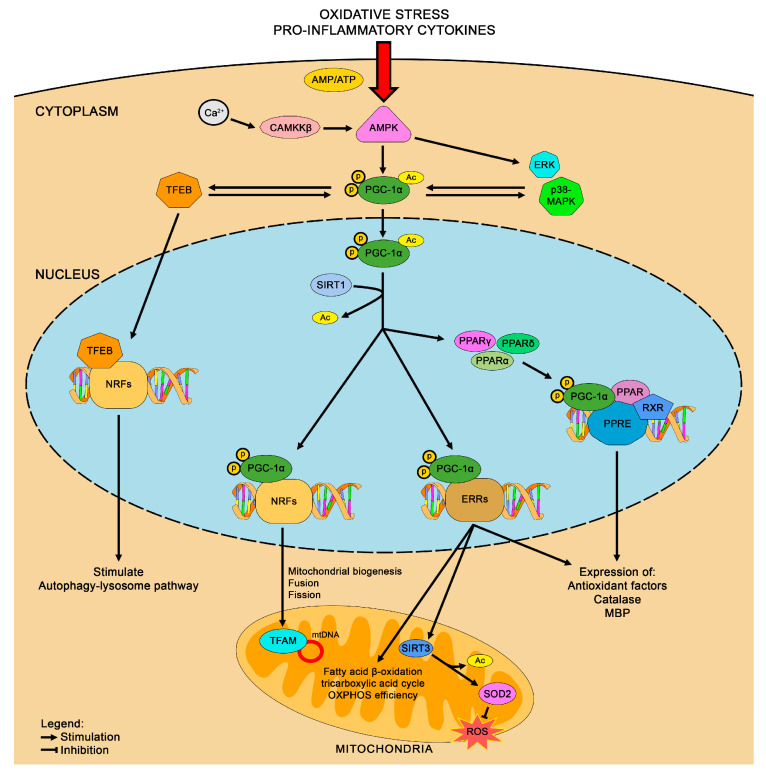
Molecular mechanisms underlying the antioxidant effects of PGC-1α signaling.

**Figure 3 life-15-00577-f003:**
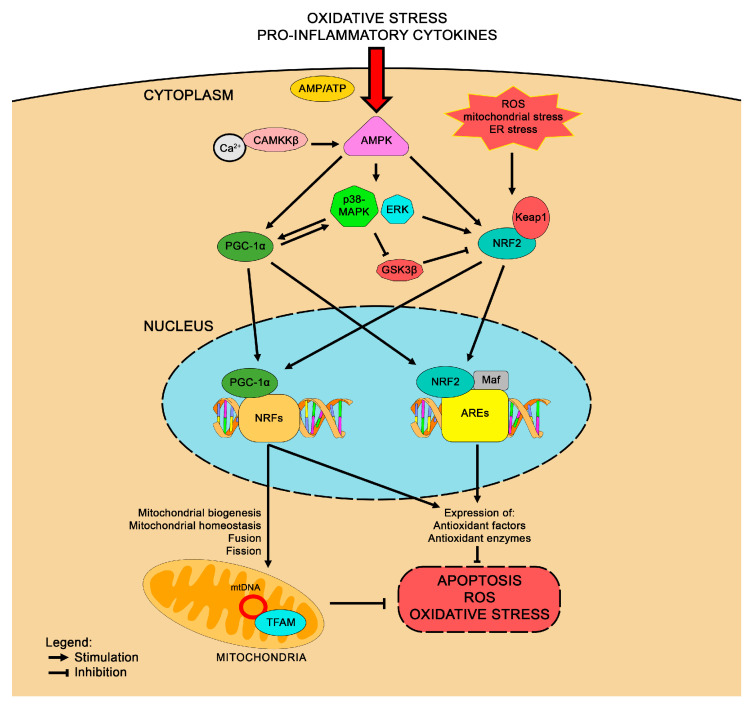
Overview on the complex molecular interactions between NRF2 and PGC-1α underlying the antioxidant effects of their signaling.

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
