# Peer review of "Antioxidant and Anti-Inflammatory Defenses in Huntington’s Disease: Roles of NRF2 and PGC-1α, and Therapeutic Strategies"

_life, 2025, doi:10.3390/life15040577_

Round 1
Reviewer 1 Report
Comments and Suggestions for Authors
The review by DEgidio et al brings forth NRF2 and PGC1A as important factors in understanding the molecular pathology of HD.
The topic is important, and the manuscript is well-written.
I wish to comment on the scheme in Fig 1. It is too cluttered and not focused. I think the manuscript will benefit from more schemes showing NRF2, PGC1A and how the two function together.
More minor comment, the translation regulation over NRF2 is important in the context of HD pathology (PMID: 24516159) and oxidative stress, and this should be discussed in the review.
Author Response
Reviewer 1
The review by DEgidio et al brings forth NRF2 and PGC1A as important factors in understanding the molecular pathology of HD.
The topic is important, and the manuscript is well-written.
Response: We would like to thank Reviewer 1 for the positive comments and the time spent reading our manuscript. We are sure that the comments provided helped in improving our manuscript. We tried to address all the points raised.
Comments
- I wish to comment on the scheme in Fig 1. It is too cluttered and not focused. I think the manuscript will benefit from more schemes showing NRF2, PGC1A and how the two function together.
Response: We would like to thank the Reviewer 1 for the suggestion, and we agree. Indeed, we provided new figures describing separately these factors and their interaction for better clarity.
- More minor comment, the translation regulation over NRF2 is important in the context of HD pathology (PMID: 24516159) and oxidative stress, and this should be discussed in the review.
Response: We thank the Reviewer for giving us the opportunity to improve our article. Indeed, we acknowledge the pivotal role of NRF2 translation regulators both in physiology and HD pathology. We discussed their involvement in NRF2 regulation in Section 3.1. Please check lines 261-275.
Reviewer 2 Report
Comments and Suggestions for Authors
The review provides a comprehensive overview of the mechanisms involved in Huntington's disease.
Here are my comments:
1] Line 11 contains a repetition of “Abstract: Huntington’s“
2] The structure and linguistic clarity of the introduction require refinement to enhance coherence
3] a.The letters in Figure 1 are not clearly visible.
b.The figure legend should be placed at the bottom.
4] The discussion section requires further elaboration to fully synthesize the findings.
5] The presentation of Table 1 should conform to the formatting and style guidelines specified by the journal.
Author Response
Reviewer 2
The review provides a comprehensive overview of the mechanisms involved in Huntington's disease.
Response: We would like to thank Reviewer 2 for the time spent reading our manuscript and for the comments provided that helped in improving our article. We tried to address all the points raised.
Comments
- Line 11 contains a repetition of “Abstract: Huntington’s“
Response: We thank the Reviewer 2 for the comment. We modified the text removing the unnecessary words. Please check the manuscript in line 11.
- The structure and linguistic clarity of the introduction require refinement to enhance coherence.
Response: We totally agree with the Reviewer. We thoroughly revised the Introduction text to improve clarity and coherence. Please check the Introduction section.
- a) The letters in Figure 1 are not clearly visible.
Response: We thank the Reviewer for the comment. We improved the text an quality of our figures.
- b) The figure legend should be placed at the bottom.
Response: We appreciate the Reviewer comment. We provided the figure legend at the bottom of the figures, left corner, in all the figures created. Please check Figures 1, 2 and 3.
- The discussion section requires further elaboration to fully synthesize the findings.
Response: We appreciate the Reviewer’s suggestion. We integrated the Discussion section further elaborating the reported findings. Please check the Discussion, particularly in lines 637-662.
- The presentation of Table 1 should conform to the formatting and style guidelines specified by the journal.
Response: We thank the Reviewer 2 for the suggestion. We now formatted the table accordingly to the specified guidelines.
Reviewer 3 Report
Comments and Suggestions for Authors
Introduction
Lack of detailed explanation on how NRF2 and PGC-1α could be activated or targeted therapeutically.
Absence of a clear hypothesis or novel insight to frame the review.
No emphasis on recent advances or the latest research in the field.
Main Body of the Manuscript
Limited discussion on clinical applicability and challenges in translating preclinical findings into human treatments.
Focus on animal models without a thorough discussion of their translation to human clinical trials.
Redundancy in the explanation of NRF2’s role in oxidative stress and inflammation.
Insufficient exploration of the side effects or risks associated with proposed therapies.
Therapeutic Strategies
Lack of specific clinical data supporting the effectiveness of therapies for HD patients.
Limited discussion on the side effects or risks of the discussed treatments.
No clear roadmap or integrated strategy for combining therapies in clinical trials.
Mention of unpublished trial results without sufficient context, leading to uncertainty.
Overly optimistic tone regarding preclinical findings, lacking a balanced view on the challenges faced in clinical trials.
Suggestions for Improvement
Provide a more detailed explanation of NRF2 and PGC-1α activation and therapeutic targeting.
Introduce a clear research question or hypothesis to give the review a more purposeful direction.
Highlight recent research or innovations to make the manuscript feel more current.
Focus on translating preclinical findings into clinical applications, addressing side effects and limitations.
Include specific examples of clinical trials and discuss their successes or challenges.
Include more clinical data and a clearer discussion of side effects and risks.
Provide a synthesized roadmap or strategy for combining therapies in clinical trials.
Avoid overly optimistic tones by balancing the preclinical results with a more realistic assessment of clinical translation.
Author Response
Reviewer 3
Introduction: Lack of detailed explanation on how NRF2 and PGC-1α could be activated or targeted therapeutically. Absence of a clear hypothesis or novel insight to frame the review. No emphasis on recent advances or the latest research in the field.
Main Body of the Manuscript: Limited discussion on clinical applicability and challenges in translating preclinical findings into human treatments. Focus on animal models without a thorough discussion of their translation to human clinical trials. Redundancy in the explanation of NRF2’s role in oxidative stress and inflammation. Insufficient exploration of the side effects or risks associated with proposed therapies.
Therapeutic Strategies: Lack of specific clinical data supporting the effectiveness of therapies for HD patients. Limited discussion on the side effects or risks of the discussed treatments. No clear roadmap or integrated strategy for combining therapies in clinical trials. Mention of unpublished trial results without sufficient context, leading to uncertainty. Overly optimistic tone regarding preclinical findings, lacking a balanced view on the challenges faced in clinical trials.
Response: We would like to thank Reviewer 3 for the in-depth comment on our article and for the valuable suggestions to improve it. We tried to carefully address all the points raised.
Comments
- Provide a more detailed explanation of NRF2 and PGC-1α activation and therapeutic targeting.
Response: We appreciate the Reviewer’s valuable suggestion. We provided more information regarding NRF2 and PGC-1α activation specific for each therapeutic compound discussed. Please check Chapter 4.
- Introduce a clear research question or hypothesis to give the review a more purposeful direction.
Response: We appreciate the Reviewer’s comment. We modified the text in the Introduction section highlighting the literature gap over the explored topic for a more purposeful direction of the manuscript. Please check lines 86-91.
- Highlight recent research or innovations to make the manuscript feel more current.
Response: We thank the Reviewer for the valuable comment. We highlighted recent research as suggested. Please check the manuscript in lines 483, 506, 520, 532, 572, 687.
- Focus on translating preclinical findings into clinical applications, addressing side effects and limitations.
Response: We thank the Reviewer 3 for the comment. Of all the cited compounds in the manuscript, a total of five clinical trials exists in HD context. However, only two of them show results. We added information regarding side effects and limitations observed across the study. Please check lines 577-582, and 599-602.
- Include specific examples of clinical trials and discuss their successes or challenges.
Response: We appreciate the Reviewer’s suggestion. Our discussion encompasses five different clinical trials, also showed in Table 1, aiming at modulating either NRF2 or PGC-1α in HD context. Of these five trials, two studies display unpublished results while one is still ongoing. The last two trials are concluded, and their results have been discussed. Please check lines 577-582, and 599-602.
- Include more clinical data and a clearer discussion of side effects and risks.
Response: We appreciate the Reviewer’s comment. We added a new clinical trial discussing its results and safety concerns. Moreover, side effects and risks have been introduced also for another clinical study. Please refer to lines 577-582, and 599-602 for additional information.
- Provide a synthesized roadmap or strategy for combining therapies in clinical trials.
Response: We thank the Reviewer for the suggestion. We described a potential roadmap for combining therapies in clinical trials in the Discussion section. Please check the manuscript in lines 714-728.
- Avoid overly optimistic tones by balancing the preclinical results with a more realistic assessment of clinical translation.
Response: We appreciate the Reviewer’s suggestion and we totally agree. Indeed, we discussed the bridge between preclinical results and clinical translation highlighting the need for careful consideration of translational limitations to realistically assess clinical applicability. Please check lines 714-737.
Round 2
Reviewer 2 Report
Comments and Suggestions for Authors
The legend of Figure 1 should be placed at the bottom. Please recheck
Reviewer 3 Report
Comments and Suggestions for Authors
Dear Authors,
Thank you for your detailed and thoughtful revisions in response to my comments. I appreciate the additional explanations on NRF2 and PGC-1α activation, as well as the clearer research direction you have incorporated. The inclusion of recent studies, expanded discussion on clinical applications, and balanced assessment of translational challenges have significantly improved the manuscript.
With these revisions, I am pleased with the changes, and I now find the manuscript suitable for publication.